# Is There Influence of Gender and the Specificity of Sports Activities on the Performance of Body Balance in Young Athletes?

**DOI:** 10.3390/ijerph192215037

**Published:** 2022-11-15

**Authors:** Michalina Czarnota, Katarzyna Walicka-Cupryś

**Affiliations:** Institute of Health Sciences, College of Medical Sciences, University of Rzeszow, 35-959 Rzeszow, Poland

**Keywords:** swimming, soccer, dancing, archery, youth, Zebris, BMI

## Abstract

Sports training can significantly influence specific motor skills. The aim of this study was to investigate the influence of gender and the specificity of sports activities on body balance, symmetry of lower extremity loads (SI) as well as body mass index (BMI) in young athletes aged 14 to 17. There were 240 participants (145 boys and 95 girls) divided into five groups: swimmers, dancers, soccer players, archery and control group. The average age was 16. Participants had 3 years of training experience (training three times a week or training that lasted between 4.5 and 6 h weekly). To assess balance, the stabilized Zebris platform was used in the study. The SI was calculated based on the percentage load on the lower extremities, dividing the greater value by the lower. Body mass index was calculated on the basis of height and weight. Women had a significantly lower BMI and balance in some analyzed parameters, while men had better SI. Archers obtained the best results of the balance parameters and the worst results of SI. Only swimmers and soccer players had a normal SI. Sports specialization generates significant differences in the manifestation of balance compared to the gender variable and sport discipline.

## 1. Introduction

The ability to maintain the center of the body mass at the base of the support is known as body balance [1]. Integration of sensory stimuli from the sensorimotor, visual, and vestibular receptors, as well as the generation of appropriate muscle synergies, enables people to achieve and maintain equilibrium under static conditions. The process of adolescence leads to individual differences in the time of growth, maturation, and development during childhood and adolescence, also influencing the results in terms of balance in adolescents [1,2]. 

The underlying developmental processes have been thoroughly investigated, and it has been shown that physical factors (e.g., weight gain and height) have only a slight influence on balance in adolescents [3,4]. The gradual maturation of the brain [2,5], the improved integration of sensory information [1,6], and the use of various postural control strategies for specific motor tasks have a significant impact on the improvement of balance results in this age group [7,8]. 

Body balance is a frequently evaluated parameter among young people [9]. It is assessed on the basis of several parameters verified with the use of a stabilometric platform that records the forces of pressure on the ground, mainly the location and projection of the center of pressure of the feet [10].

The device also allows for the percentage determination of the degree of load between the limbs and the method of loading the forefoot and hindfoot of each foot. Little information is available in the literature on the symmetry of loading in the lower extremities [11,12], although it is an important parameter that defines the symmetry or asymmetry in the way the lower extremities are loaded. Quantitative indicators of body stability, variability, and amplitude of the foot pressure point on the ground distinguish young people during physical activity [7]. The main difference in the way of regulating the balance of standing posture is the speed of displacement of the pressure point with the feet on the ground. It may be reflected in the efficiency of the central nervous system [8]. 

Adolescence is also the time when children usually train in a chosen sport. Sport, in addition to developing motor skills and being fun, makes it possible to develop teamwork skills, builds bonds among peers, and has an impact on self-esteem [13]. The American Pediatric Academy Committee for Sports Medicine and Fitness of the American Pediatric Academy recommends playing one sport activity up to five days a week with at least one day without physical activity [14]. 

The adolescence period is one in which intense changes take place in the human body: sexual maturation, changes in body composition, and rapid skeletal growth [15]. The onset and rate of progression of pubertal events vary between adolescents, but pubertal changes occur in a predictable stepwise manner [16,17,18]. Implementing balance training (in addition to agility training and eye-hand coordination training) before adolescence was shown to lead to increased activation and adaptation of neurons (increased synaptoplasticity) [19].

In the literature, studies are conducted that evaluate the impact of BMI on the balance of young people [20]. However, the results of these studies are not conclusive and require further interventions. There is preliminary evidence that BMI is correlated with the variability of posture movement in standing posture [21,22,23]. Many authors have analyzed the body mass index and its impact on body balance, but the results of the authors’ research differ in terms of their results. Scientists evaluated the relationship between body mass index (BMI) factors and balance parameters in children and adolescents [24]. The study group consisted of 1337 children aged 7 to 15. There were 559 girls and 578 boys among the subjects. It turned out that overweight and obese children have better stability parameters than children with normal body weight. This is also confirmed in many studies [22,25]. On the other hand, there are also studies that do not agree with these results [11,12,21,23,26,27]. However, body posture is obtained at the age of about 15 years [28,29,30]. In one study, young people who did not practice any sports discipline were investigated. The authors of this study attempted to check whether three summer training sessions in popular sports affect the balance, symmetry of loading on the lower limbs and the BMI index. Physical activity in various sports has a different character and may, to a different degree, influence the formation of selected motor features [31,32,33,34]. 

The analysis of the literature ambiguously determines gender dominance in the field of youth balance research [35,36,37]. However, according to research, the posturographic analysis indicates differences in postural control between genders. Men have higher stability, with lower oscillations and speeds of center of pressure [38]. On the other hand, different results were obtained by other researchers providing an overview of the neuromuscular differences between men and women [39]. The authors’ studies have demonstrated that female athletes have decreased proprioception, compensatory neuromuscular control patterns and enhanced static balance compared with male athletes. There is scientific evidence for the effect of short-term sports training on balance in adolescents [40,41]. There are not many studies that allow the assessment of balance among adolescents practicing various sports, including comparing symmetrical and asymmetrical sports. 

This paper attempts to identify how gender and the inclusion of students in different sports activities (team sports, individual sports) generate differences in performance on static balance tests and BMI:(1)Does gender matter in the balance, symmetry index and BMI between young athletes practicing various sports?(2)Is there a difference between the balance, symmetry index and BMI among young athletes practicing various sports and not training?

## 2. Materials and Methods

### 2.1. Participants

There were 240 participants in the study (145 boys and 95 girls) aged 14 to 17. They were divided into 5 groups, 48 people each. Group I consisted of swimmers, group II dancers, people in group III played soccer and in group IV train archery. There was a control group included people who did not play any sport. Participants in groups I–IV were members of a sports club in the Subcarpathia province. 

The participants had a three-year training experience. During this time, the training was conducted three times a week or lasted between 4.5 and 6 h a week. 

### 2.2. Data Collection

Data collection was carried out between October 2014 and April 2015. To assess the balance of the participants, measurements were made using FDMS Zebris stabilometric platform. The study was conducted once. This method was considered reproducible [10,42]. The device consists of individually calibrated capacitive force sensors. The static module of software allows for the analysis of the distribution of the pressure of the foot on the ground when a patient is freely standing, as well as the assessment of the balance thanks to the parameters of the location of the projection of the center of the pressure of the foot on the platform and the asymmetry of setting and load of the feet (right–left, front–back). Examination of each child consisted of maintaining a standing position on the platform for 20 s. The examined person stood freely, without shoes, with the upper limbs lowered and his eyes focused on the wall at eye level and with feet hip-width apart. They were asked not to move their upper limbs or heads during examination. As a result of the study, the following parameters were received: ellipse area as shown by projection of the center of force of the feet on the platform (CEArea) [mm^2^], distance covered by projection of the center of force of the feet on the platform (COF TTL) [mm], lateral inclination of projection of the center of force of the feet on the platform (COF HD) [mm], anterior–posterior projection of the center of force of the feet on the platform (COF VD) [mm]. To describe the difference between the left and right loadings symmetry index (SI) was calculated. The percentage load on the lower limbs obtained from the balance report was calculated according to the weight of each subject. In the next step, the greater value of the load was divided by the lower one. The correct range for the symmetry index is between 1.00 and 1.15 [43]. The body mass index was calculated on the basis of height and weight. Each participant was measured standing upright, wearing underwear, and no shoes on medical electronic personal scales with a height indicator RADWAG C315. 

### 2.3. Ethical Considerations

The research was approved by the Bioethics Committee of the University of Rzeszów (File No. 2014/17). Due to the age of the respondents, the free and informed consent of a parent or legal guardian was obtained from all participants after being invited to participate in the study, explained the objectives and guaranteed anonymity and confidence of the data, through the attribution of a sequential numerical code to each participant, which was used to identify the participants. 

### 2.4. Data Analysis

Statistical analysis of the material was performed using Statistica 13.1 software from StatSoft. To compare the results of the measurements taken in the four groups, the one-factor variance analysis Anova was used, or, in cases when the assumptions of the normality of the distributions of given variables were not met, the Kruskal–Wallis Anova test was used. The post hoc tests for the mentioned tests were Tukey’s test and multiple comparison test as well as the Dunn’s multiple comparison test for pairwise comparison, made to check exactly which groups differ from each other. The test results are presented as adjusted and unadjusted data controlled using Holm’s progressive step-up procedure. The relations between qualitative variables were found using the χ² Pearson’s test and the relationships between quantitative variables were found using the Spearman’s rank correlation test. The level of statistical significance was established at *p* < 0.05.

## 3. Results

### 3.1. Clinical Characteristics of the Participants

Participants were aged 14 to 17, and the average age was 16. The difference in age between the groups was statistically significant (*p* < 0.001). The 16-year-olds were the largest group among swimmers, dancers were mostly 16 or 17. The 15-, 16- or 17-year-old age groups were more or less equal between soccer players as well as archers, and the 17- and 15-year-old age groups were the most common in the control group (Table 1).

There were 60.4% men and 39.6% women among the participants (Table 2). Only men played soccer, and there were equal numbers of men and women dancing. There were a few more women than men among swimmers and a few more men than women in the archer and control groups. The differences in the distribution of the participants of different sexes in the groups were statistically significant (*p* < 0.001).

### 3.2. Balance Results

Significant differences were found in the values of body mass index, balance parameters and symmetry index between women and men practicing various sports, as shown in Table 3. Women were characterized by a significantly lower body mass index and balance in some analyzed parameters, while men were characterized by better symmetry indexes.

Significant differences were observed between the COF TTL (*p* = 0.031) and CEArea (*p* = 0.028) in the archery group and all other groups. Archers had the best CEArea values. There were no statistically significant differences between the values of the COF HD (*p* = 0.432) and COF VD (*p* = 0.211) parameter in all five groups, but the archery group had the best COF HD results, and the swimming group had the best COF VD results (Table 4).

The results of statistical significance for the COF TTL parameter were confirmed by Dunn’s test (Table 5).

The symmetry index was calculated as the ratio of the greater pressure of one of the sides of the body to the lower pressure (Table 6). There were observed statistically significant differences between the values of the SI of archery group and other groups (*p* = 0.004). The archery group had the worst value, and only the swimming and soccer group had a normal value.

Dunn’s test showed significance between the archers and soccer groups (Table 7).

Differences in BMI among the five groups were significant (*p* < 0.001). Significant differences were observed between the BMI of the dancers and all other groups. Dancers had the lowest mean BMI. The mean BMI indices in other groups were similar (Table 8). These results were confirmed by Dunn’s test (Table 9).

## 4. Discussion

Standing posture control is one of the most important human motor behaviors and, at the same time, the starting point for other motor activities. Balance is a complicated function that involves several different body systems, the vestibular system, vision, the proprioceptor, and the muscle group. It is the ability to retain the center of mass above the base of support when stationary (static balance) or moving (dynamic balance) [43,44].

Schedler et al. performed a meta-analysis that pointed out that the development of balance does not end in childhood but lasts throughout adolescence [45]. Adolescence is an intense period in the development of an organism that differs in terms of gender. In balance tests, women aged 14–16 generally perform better than men; however, they do not show improvement in motor performance after age 14 years, while men continue to improve throughout puberty [46,47,48,49]. 

The aim of this study was to investigate whether gender matters in the balance, symmetry index and BMI between young athletes practicing various sports. Research in this area seems to indicate an advantage of women over men. The postural balance of girls who practice alpine skiing is superior to boys of the same age (13 years) in the mid-lateral plane [50]. A comparison of static balance results for young athletes in Slovenia identifies higher values of dancers compared to other sports, and girls perform significantly better than boys [51]. Our research indicates superior female performance for two of the four parameters determining the static body balance (Table 3). In the other two parameters, no statistical significance was noted; however, the COF VD was close to statistical significance *p* = 0.084 in favor of men (Table 3). 

Significant differences were also seen for BMI. Women obtained statistically significantly lower BMIs than men. Some studies have documented differences in weight-related health behaviors by gender expression within the sexes. For instance, in the Growing Up Today Study (GUTS), a national cohort of US youths, researchers found that gender-conforming adolescence girls were more likely to endorse trying to look like people in the media than gender-nonconforming girls, and gender-conforming boys were found to be more involved in sports than gender-nonconforming boys [52,53]. It was also noticed that lower values of the symmetry coefficient were associated with a higher content of all body components in children in adolescence [23]. Our own research confirms these reports. 

The purpose of this study was also to evaluate the ability to maintain balance between adolescents practicing swimming, dancing, football and archery between adolescents who do not. The results of the research showed that better results of balance belong to the group of archers (Table 4)—they obtained statistically significantly better results in two of the four parameters determining the balance. Recreational archery is a very safe sport [54]. However, based on the conducted research, it can be presumed that the conditions specific to the archery being practiced may modify the natural strategies of posture in order to maintain the best possible results in a static position. Taking into account the fact that the balance test was performed in a position different from the position in which specialization training is carried out, it can be assumed that the repetitive body position with disproportionate load on the lower limbs, ensuring better stability, may be remembered by the body and used in everyday life. These results are confirmed in research, where scientists compared the effects of archery and taekwondo experience on bipedal postural control [31]. On the basis of the obtained results, they suggest that the shooting experience has a greater impact on the reduction in rolling under static equilibrium conditions.

The second group in terms of high balance parameters are swimmers. Although no statistically significant differences were found, the results of their studies stand out from the rest of the groups. The issue of balance among swimmers has been researched several times. There is research that compared the balance control of young swimmers and judokas (aged 10.5 to 17) [33]. The objective of the study was to verify the basic mechanism of balance control in judokas compared to swimmers (no balance training). The results show that judokas cope better than swimmers cope temporary sight loss and compensate for unstable ground to a lesser extent. The results suggest that judokas, compared to swimmers, are less visually dependent on their sensorimotor system. Baccouch studied young people (aged 11 to 13 years) who trained in swimming and kung fu [34]. The study resulted in the opinion that both types of sports can be recommended to adolescents as recreational or rehabilitation activities due to their potential to develop balance.

In this study, the dancers group did not obtain significant results; they were comparable with the control group. Some research shows the impact of ballet on balance [46,47,49,55]. Hutt and Redding’s studies have shown that dancing helps develop coordination skills that affect improved dynamic balance [55]. However, the researchers proved the positive impact of various forms of dance on balance and attributed the effect of better results to the effect of eyesight on maintaining balance [47,49]. Hugel et al. proved that classical ballet training develops specific modalities of balance. Visual input is important because dancers showed better results than controls with open eyes and only under open conditions [46]. The balance results for the dance group training studied did not confirm the reports of other authors. The group of dancers did not obtain the best results in any of the four parameters describing the balance; moreover, in two of them, the group obtained the worst results. 

The symmetry of load of the lower limbs is another important method, in addition to balance, to develop posture control and moving patterns. The study results show that only soccer players and swimmers had a normal level of this parameter. The proper distribution of ground reaction forces ensures the maximum effectiveness of the load and the correct biomechanics of the lower limb [56]. The results obtained by archers seem to be disturbing, as they differ significantly from the assumed norm. There is a probability that the obtained results may be influenced by the change in loads during training, which allows one to obtain high balance results. This may suggest that after the end of training, the body remains in changed conditions and consolidates them during subsequent training, as well as during everyday life. Shifting a much greater body weight to one side may result in future overload changes on the overloaded side, injuries and, consequently, premature degenerative changes.

## 5. Conclusions

The specificity of sports activities and gender have influence on the performance of body balance in young athletes. Sports specialization generates significant differences in the manifestation of balance compared to the gender variable—supporting the superiority of women in terms of balance tests and BMI. 

According to the practiced sport, the performances obtained indicate the level of demand for static stability.

For a more accurate understanding of the influence of gender on weight in adolescence athletes, it is recommended to conduct tests in a larger group of adolescents and in a long-term manner using a body mass composition analyzer to exclude high values of the BMI index caused by muscle mass. 

## Figures and Tables

**Table 1 ijerph-19-15037-t001:** Age of subjects classified by discipline.

Age	Swimming		Dancing		Soccer		Archery		Control Group		Participants	
	N	%	N	%	N	%	N	%	N	%	N	%
14 y/o	8	16.7	7	14.6	0	0.0	1	2.1	0	0.0	16	6.7
15 y/o	3	6.3	5	10.4	13	27.1	11	22.9	14	29.2	46	19.2
16 y/o	36	75.0	23	47.9	17	35.4	13	27.1	3	6.3	92	38.3
17 y/o	1	2.1	13	27.1	18	37.5	23	47.9	31	64.6	86	35.8
Total	48	100.0	48	100.0	48	100.0	48	100.0	48	100.0	240	100.0
*p*					χ²(9) = 82.31 *p* < 0.001							

N—number, χ²—value of χ² Pearson’s test; *p*—indicator of the testing probability of testing.

**Table 2 ijerph-19-15037-t002:** Distribution of participants and related percentages according to gender and specific sports activity.

Gander	Swimming		Dancing		Soccer		Archery		Control Group		Participants	
	N	%	N	%	N	%	N	%	N	%	N	%
Women	27	11.2	24	10.0	0	0.0	22	9.2	22	9.2	95	39.6
Men	21	8.8	24	10.0	48	20.0	26	10.8	26	10.8	145	60.4
Total	48	20.0	48	20.0	48	20.0	48	20.0	48	20.0	240	100.0
*p*		χ²(3) = 40.38 *p* < 0.001										

χ²—value of χ² Pearson’s test; *p*—indicator of the testing probability of testing.

**Table 3 ijerph-19-15037-t003:** Comparison of results of BMI, balance and symmetry index measurements in young athletes practicing various sports by gender.

Variable	Females (N = 73)		Males (N = 119)		
	Mean	SD	Mean	SD	*p*
BMI	20.12	2.91	21.84	3.11	**0.003**
COF TTL (mm)	734.81	142.6	896.96	173.7	**<0.001**
COF HD (mm)	2.35	1.51	2.52	1.7	0.126
COF VD (mm)	4.20	2.12	3.86	2.26	0.084
CEArea (mm^2^)	52.30	34.91	98.73	62.32	**<0.001**
SI	1.20	0.38	1.13	0.21	**<0.001**

BMI [kg/m2]— body mass index; COF TTL [mm]—distance covered by the projection of the center of force of the feet onto the platform; COF HD [mm]—lateral inclination of the projection of the center of force of the feet onto the platform; COF VD [mm]—anterior–posterior projection of the center of force of the feet onto the platform; CEArea [mm2]—area of the ellipse as shown by the projection of the center of force of the feet onto the platform; SI—symmetry index; mean—average value; p—statistical significance; SD—standard deviation; N—number; bold number means a statistically significant result.

**Table 4 ijerph-19-15037-t004:** Comparison of results of balance parameters depending on the practiced sport.

		Basic Statistics				
		Swimming	Dancing	Soccer	Archery	Control Group
COF TTL (mm)	Number	48	48	48	48	48
	Median	882.9	852.0	856.5	750.6	825.2
	IR	798.6–972.2	764.1–1040.2	747.1–963.9	716.8–914.2	722.5–1029.1
	*p*	H = 2.18 *p* = 0.031				
COF HD (mm)	Number	48	48	48	48	48
	Median	2.60	2.60	2.75	2.30	2.60
	IR	1.95–3.25	1.80–3.50	1.90–3.60	1.24–3.11	1.55–3.85
	*p*	H = 0.36 *p* = 0.432				
COF VD (mm)	Number	48	48	48	48	48
	Median	3.70	4.10	4.10	3.65	4.05
	IR	2.80–4.90	3.30–5.65	3.00–5.20	2.90–4.95	2.60–6.05
	*p*	H = 3.79 *p* = 0.211				
CEArea (mm^2^)	Number	48	48	48	48	48
	Median	46.05	50.65	45.65	35.40	40.65
	IR	25.60–65.65	36.05–80.90	24.60–66.90	20.70–62.85	21.15–88.50
	*p*	H = 1.09 *p* = 0.028				

COF TTL [mm]—distance covered by the projection of the center of force of the feet onto the platform; COF HD [mm]—lateral inclination of the projection of the center of force of the feet onto the platform; COF VD [mm]—anterior–posterior projection of the center of force of the feet onto the platform; CEArea [mm^2^]—area of the ellipse as shown by the projection of the center of force of the feet onto the platform; H—value of Kruskal–Wallis Anova test; *p*—indicator of the probability of the test; IR—interquartile range.

**Table 5 ijerph-19-15037-t005:** Dunn’s Test for multiple comparisons for the path of the center of gravity (COF TTL).

Comparison	Z	Unadjusted *p*-Values	Adjusted *p*-Values
Control group–archery	3.04160601	2.35 × 10^−3^	1.65 × 10^−2^
Control group–swimming	−1.10109372	2.71 × 10^−1^	1.00
Control group–soccer	−0.11540168	9.08 × 10^−1^	1.00
Control group–dancing	−1.14519627	2.52 × 10^−1^	1.00
Archery–swimming	−4.14269972	3.43 × 10^−5^	3.09 × 10^−4^
Archery–soccer	−3.15700769	1.59 × 10^−3^	1.28 × 10^−2^
Archery–dancing	−4.18680228	2.83 × 10^−3^	2.83 × 10^−4^
Swimming–soccer	0.98569204	3.24 × 10^−1^	9.73 × 10^−1^
Swimming–dancing	−0.04410255	9.65 × 10^−1^	9.65 × 10^−1^
Soccer–dancing	−1.02979459	3.03 × 10^−1^	1.00

Z—values for the Z test statistic for each comparison; *p*-values adjusted with the Holm method.

**Table 6 ijerph-19-15037-t006:** Symmetry index (SI) depending on the practiced sport.

SI	Basic Statistics		
	Number	Median	Interquartile Range
swimming	48	1.13	1.05–1.21
dancing	48	1.13	1.05–1.27
soccer	48	1.08	1.04–1.14
archery	48	1.48	1.16–1.36
control group	48	1.12	1.06–1.21
*p*	H = 4.67 *p* = 0.004		

SI—symmetry index; H—value of Kruskal–Wallis Anova test. *p*—indicator of the probability of the test.

**Table 7 ijerph-19-15037-t007:** Dunn’s Test for multiple comparisons for symmetry index (SI).

Comparison	Z	Unadjusted *p*-Values	Adjusted *p*-Values
Control group–archery	−1.66353809	9.62 × 10^−2^	4.81 × 10^−1^
Control group–swimming	0.43371077	6.64 × 10^−1^	1.00
Control group–soccer	1.92229434	5.46 × 10^−2^	4.37 × 10^−1^
Control group–dancing	0.05733803	9.54 × 10^−1^	9.54 × 10^−1^
Archery–swimming	2.09724886	3.60 × 10^−2^	3.24 × 10^−1^
Archery–soccer	3.58583243	3.36 × 10^−4^	3.36 × 10^−3^
Archery–dancing	1.72087612	8.53 × 10^−2^	5.12 × 10^−1^
Swimming–soccer	1.48858357	1.37 × 10^−1^	5.46 × 10^−1^
Swimming–dancing	−0.37637274	7.07 × 10^−1^	1.00
Soccer–dancing	−1.86495631	6.22 × 10^−2^	4.35 × 10^−1^

Z—values for the Z test statistic for each comparison; *p*-values adjusted with the Holm method.

**Table 8 ijerph-19-15037-t008:** BMI depending on the practiced sport.

BMI (kg/m^2^)		Basic Statistics			
	Number	Median		Interquartile Range	
Swimming	48	21.24		20.58–21.89	
Dancing	48	20.23		18.75–21.33	
Soccer	48	21.63		21.21–22.37	
Archery	48	21.56		20.98–21.68	
Control group	48	21.43		20.42–21.94	
*p*		H = 26.86 *p* < 0.001			
	Swimming	Dancing	Soccer	Archery	Control Group
Swimming		0.023	0.149	0.124	1.000
Dancing	0.023		<0.001	0.014	0.010
Soccer	0.149	<0.001		0.097	0.278
Archery	0.094	0.014	0.283		0.834
Control group	1.000	0.010	0.278	0.127	

BMI [kg/m^2^]—body mass index; H—value of the Kruskal–Wallis Anova test. *p*—indicator of the probability of the test.

**Table 9 ijerph-19-15037-t009:** Dunn’s Test for multiple comparisons for BMI.

Comparison	Z	Unadjusted *p*-Values	Adjusted *p*-Values
Control group–archery	−0.78429770	4.33 × 10^−1^	8.66 × 10^−1^
Control group–swimming	0.03528237	9.72 × 10^−1^	9.72 × 10^−1^
Control group–soccer	−1.84644408	6.48 × 10^−2^	3.24 × 10^−1^
Control group–dancing	3.09896824	1.94 × 10^−3^	1.55 × 10^−2^
Archery–swimming	0.81958007	4.12 × 10^−1^	1.00
Archery–soccer	−1.06214637	2.88 × 10^−1^	1.00
Archery–dancing	3.88326595	1.03 × 10^−4^	9.28 × 10^−4^
Swimming–soccer	−1.88172645	5.99 × 10^−2^	3.59 × 10^−1^
Swimming–dancing	3.06368587	2.19 × 10^−3^	1.53 × 10^−2^
Soccer–dancing	4.94541232	7.60 × 10^−7^	7.60 × 10^−6^

Z—values for the Z test statistic for each comparison; *p*-values adjusted with the Holm method.

## Data Availability

Not applicable.

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
