# Peer review of "Is There Influence of Gender and the Specificity of Sports Activities on the Performance of Body Balance in Young Athletes?"

_ijerph, 2022, doi:10.3390/ijerph192215037_

Round 1
Reviewer 1 Report (New Reviewer)
Title: Is there influence of gender and the specificity of sports activi-2 ties on the performance of body balance in young athletes?
After I review this study, I have some comments and suggestions as follows:
Abstract:
Line 9-10: this is not a prospective study; the study did not design to follow-up children individual and measure the change of body balance. The research design should to revise.
Introduction
Line 28: What is inter-idividual? Individual?
Materials and Methods
Line 99: this study did not design as the prospective study. The research design should to revise.
Line 119-120: How many times of balance assessment? If only 1 time please identify When? before or after the children received 3 years of training.
Line 153-159: Choose the appropriate statistic used for data analysis. I suggest to consult statistician and revise the statistic used such as the one-way ANOVA for more than 2 group comparison when the normality is assumed. The post hoc test is the Dunn’s multiple comparison test for pairwise comparison test. And the Pearson’s correlation test (r) is used to determine the relationship among the quantitative variables and the Spearman’s rank correlation test (rs) is used for the relationship among the qualitative variables.
Results
The author should revise table and select the appropriate statistics used for data analysis.
Line 162-163: if there are the difference in age and gender between groups (p<0.001), the difference of balance parameters (COFTTL, COFHD, COFVD, CEA, SI) among 4 type of sports and control group is confounded by age and gender. Suggest to reanalysis and reduce the effect of age and gender for balance comparison among 5 groups.
Table1: the symbol ‘c2 is the Chi-square test, it is not the Pearson’s test. please check the statistic used for data analysis. Revise table1 to be clearly present which pair is significant difference?
Table4: For the Kruskal-Wallis ANOVA test, the result will report the significant difference among groups and to identify pairwise comparison need to use the Dunn’s test. please check the result and statistic used.
I didn’t found table 8.
Suggest to revise data in table 4-10 and select only median and interquartile range (not the 1st and 3rd quartiles) to report in the table. These statistical parameters are data representing for nonparametric statistics.
Author Response
Abstract:
Line 9-10: this is not a prospective study; the study did not design to follow-up children individual and measure the change of body balance. The research design should to revise.
We revise research and made changes in text.
Introduction
Line 28: What is inter-idividual? Individual?
Linguistic mistake. It should be individual.
Materials and Methods
Line 99: this study did not design as the prospective study. The research design should to revise.
We revise research and made changes in text.
Line 119-120: How many times of balance assessment? If only 1 time please identify When? before or after the children received 3 years of training.
The study was conducted once, the condition for participation in the study was 3 years of experience.
Line 153-159: Choose the appropriate statistic used for data analysis. I suggest to consult statistician and revise the statistic used such as the one-way ANOVA for more than 2 group comparison when the normality is assumed. The post hoc test is the Dunn’s multiple comparison test for pairwise comparison test. And the Pearson’s correlation test (r) is used to determine the relationship among the quantitative variables and the Spearman’s rank correlation test (rs) is used for the relationship among the qualitative variables.
We revise research.
Results
The author should revise table and select the appropriate statistics used for data analysis.
Line 162-163: if there are the difference in age and gender between groups (p<0.001), the difference of balance parameters (COFTTL, COFHD, COFVD, CEA, SI) among 4 type of sports and control group is confounded by age and gender. Suggest to reanalysis and reduce the effect of age and gender for balance comparison among 5 groups.
The authors of the study designed studies to investigate the influence of gender and the specificity of sports activities on body balance. Based on the literature analysis, 4 different sports activities were selected for comparison and control group. One of the sports was football, represented only by men. Therefore, the analysis showed differences in the sexes between the groups. No differences were identified in the remaining groups.
Table1: the symbol ‘c2 is the Chi-square test, it is not the Pearson’s test. please check the statistic used for data analysis. Revise table1 to be clearly present which pair is significant difference?
To analyse relations between qualitative variables the χ² Pearson’s test was used. In Table 1 we can’t clearly present which pair is significant difference. It would be possible if the age was given in years (as a number), but it’s not.
Table4: For the Kruskal-Wallis ANOVA test, the result will report the significant difference among groups and to identify pairwise comparison need to use the Dunn’s test. please check the result and statistic used.
We checked the results and add Dunn’s test as suggested.
I didn’t found table 8.
We checked and revised. There was an error in the numbering of the tables.
Suggest to revise data in table 4-10 and select only median and interquartile range (not the 1st and 3rd quartiles) to report in the table. These statistical parameters are data representing for nonparametric statistics.
We revised data in the table 4-10 and select only median and interquartile range as suggested. We add Dunn’s test.
Reviewer 2 Report (New Reviewer)
Dear authors,
thank you for the paper.
You deal with a important factor of sport performance. However, the aim is not quite clear, i.e. there are many factors that influence the investigated parameters and the investigations are quite different. So please stress, what exactly is the benefit of your science apart from some significant differences between sports.
The factors which may influence the results should be worked out. I
Author Response
You deal with a important factor of sport performance. However, the aim is not quite clear, i.e. there are many factors that influence the investigated parameters and the investigations are quite different. So please stress, what exactly is the benefit of your science apart from some significant differences between sports.
The factors which may influence the results should be worked out.
In the discussion were added information regarding to factors which may influence the results.
The aim of this study was assessment of balance, symmetry index and BMI among young athletes practicing various sports and not training. A factor common to all groups was three years of experience in training a selected sports discipline. The authors of the study tried to assess the above aspects taking into account sports experience in order to draw conclusions about the possible impact of the three-year specialization on the body of young athletes. As in the case of the group of archers, their sport specialization and the nature of the discipline practiced probably contributed to their better balance results compared to other groups. However the same group, showed the greatest disproportion in loading the lower limbs, despite the fact that the test was performed in a position different from the position taken during training or competition. This allows the authors of the research to draw conclusions that young people training archery in a special way should be under the observation of the medical staff, which will periodically analyze the symmetry of loading the lower limbs and, if necessary, implement preventive programs.
Reviewer 3 Report (New Reviewer)
Dear authors,
This is an interesting article, and overall is well-written. However, the rationale for this study is unclear, and some parts of the doc lack additional explanations. Please see the below point-by-point.
Abstract
Lines 17-18 – what the authors aim to say with "superiority women in (..) BMI"? Higher BMI values in women when compared with men? Also, isn't this expected? I would remove this information since it is not relevant.
Line 18 – "Archery improves balance; this is a cross-sectional study, which means that the authors cannot establish a cause-effect. This needs to be changed.
Line 19 -this is a very speculative sentence and a very simplistic vision of motor performance and the several factors that determined them. Moreover, this ..
Introduction.
Overall, the introduction needs to be revised since is missing a clear identification of pertinent gaps in the literature related to balance that supports the study's aims. Also, it is not clear the reasons to study, within the same paper, the impact of several factors on BMI (several of them already well known). The final questions need to be revised according to pertinent information that the current literature does not respond.
Lines 46 -47 – It is missing references that support this sentence.
Lines 47 – 50 – although COP velocity parameters are sensitive measures, there are other important measures to consider depending on the study aim. The authors need to present references that support the sentence.
Line 64- the relevance of the rationale starts in this line. The previous text does not show any connection with this paragraph.
Lines 28-30 – this sentence needs evidence. The studies presented are not properly saying that mature body posture is reached at 15 yrs. The authors need to revise this.
Design
Lines 102 – 107 - Information on pilot studies that have not been published seems unnecessary. It will confuse the readers.
Data collection
How many trials were performed by each subject? Were the feet together or separated by a certain distance?
Please explain better how the symmetry index is computed.
Discussion
Line 252- the authors have twice "for two" in the sentence.
Again, discussing differences between boys and girls regarding BMI is not adding any relevant information to the current literature. Concerning sports, the authors need to discuss with caution the impact of different sports on balance since this a cross-sectional study and many other factors are for sure contributing to balance. In addition, balance measures are typically characterized by variability, which increases the need for caution regarding data interpretation.
Conclusions
"Gender differences support a superiority of women in balance test and BMI."- again, what do the authors intend to say with this sentence?
"Archery improves balance, but may have an adverse effect on the symmetry of load of the lower limbs"- the present study does not allow this conclusion.
Author Response
Abstract
Lines 17-18 – what the authors aim to say with "superiority women in (..) BMI"? Higher BMI values in women when compared with men? Also, isn't this expected? I would remove this information since it is not relevant.
Women had a significantly lower BMI and balance in some analyzed parameters, while men had better SI. We revised.
Line 18 – "Archery improves balance; this is a cross-sectional study, which means that the authors cannot establish a cause-effect. This needs to be changed.
We revised.
Line 19 -this is a very speculative sentence and a very simplistic vision of motor performance and the several factors that determined them. Moreover, this ..
We revised.
Introduction.
Overall, the introduction needs to be revised since is missing a clear identification of pertinent gaps in the literature related to balance that supports the study's aims. Also, it is not clear the reasons to study, within the same paper, the impact of several factors on BMI (several of them already well known). The final questions need to be revised according to pertinent information that the current literature does not respond.
We revised paper. Many scientists have analyzed the body mass index and its impact on body balance, but the results of the authors' research differ in terms of their results. Therefore, the authors of this study attempted an analysis taking into account BMI, as the study group (48 people in the same age group with three years of training experience) is a valuable research sample).
Lines 46 -47 – It is missing references that support this sentence.
We revised.
Lines 47 – 50 – although COP velocity parameters are sensitive measures, there are other important measures to consider depending on the study aim. The authors need to present references that support the sentence.
We revised.
Line 64- the relevance of the rationale starts in this line. The previous text does not show any connection with this paragraph.
The authors of the study wanted to emphasize the significance of adolescence and to draw attention to other changes taking place in the body at that time, which are a physiological consequence of the development of the body. If it would not be necessary we can remove this paragraph.
Lines 28-30 – this sentence needs evidence. The studies presented are not properly saying that mature body posture is reached at 15 yrs. The authors need to revise this.
We revised.
Design
Lines 102 – 107 - Information on pilot studies that have not been published seems unnecessary. It will confuse the readers.
We removed this information.
Data collection
How many trials were performed by each subject? Were the feet together or separated by a certain distance?
Please explain better how the symmetry index is computed.
The examined person stood freely, without shoes, with the upper limbs lowered and his eyes focused on the wall at eye level and with feet hip-width apart. We revised.
To assess the symmetry of load of the lower limbs symmetry index (SI) was calculated. The percentage load on the lower limbs obtained from the equilibrium report was calculated according to the weight of each subject. In the next step, the greater value of the load was divided by the lower one.
Discussion
Line 252- the authors have twice "for two" in the sentence.
We revised.
Again, discussing differences between boys and girls regarding BMI is not adding any relevant information to the current literature. Concerning sports, the authors need to discuss with caution the impact of different sports on balance since this a cross-sectional study and many other factors are for sure contributing to balance. In addition, balance measures are typically characterized by variability, which increases the need for caution regarding data interpretation.
Thank you for your valuable suggestions, we've made some corrections.
Conclusions
"Gender differences support a superiority of women in balance test and BMI."- again, what do the authors intend to say with this sentence?
We revised.
"Archery improves balance, but may have an adverse effect on the symmetry of load of the lower limbs"- the present study does not allow this conclusion.
We revised.
Round 2
Reviewer 1 Report (New Reviewer)
Thank you for submit the revised version, I have some suggestions that I think it will improve your paper. Revise table 4-12;
1. select parameter to present your data; mean and SD or median and IQR which one more appropriate to present your data?
2. too many tables and I suggest to reduce the table by combining data presentation and pairwise comparison.
3. What is the unadjusted and adjusted p-value? Need to explain in the statistical analysis section.
4. Check the reference style (Ref No. 10, 49, 55, 57)
Author Response
Dear Reviewer,
Below I'm sending answers to your comments. Thank you for your valuable suggestions.
- select parameter to present your data; mean and SD or median and IQR which one more appropriate to present your data?
I our opinion median and interquartile range is more appropriate. The selected data is the basis for the Dann test which is also presented. We revised table 4-12.
- too many tables and I suggest to reduce the table by combining data presentation and pairwise comparison.
We revised data. The data describing the balance are presented in one table.
- What is the unadjusted and adjusted p-value? Need to explain in the statistical analysis section.
We add explain in statistical analysis section: The post hoc tests for the mentioned tests were Tukey’s test and multiple comparison test as well as the Dunn’s multiple comparison test for pairwise comparison, made to check exactly which groups differ from each other. The test results are presented as adjusted and unadjusted data controlled using Holm's progressive step-up procedure.
- Check the reference style (Ref No. 10, 49, 55, 57).
We change the reference style.
With Kind Regards
Reviewer 2 Report (New Reviewer)
Thank you for the improvements of contents etc.. The publication is clearly presented and discussed and can be publish now.
Author Response
Dear Reviewer,
Thank you for your support and valuable tips that allowed us to improve of contents.
Reviewer 3 Report (New Reviewer)
The authors made an effort to improve the manuscript, but some of the initial issues remained in this current version. Please see the following comments:
Abstract – As previously mentioned, the conclusions are speculative since the experimental design does not allow such conclusions. Therefore, the authors need to revise this.
Line 49 – as far as I could understand, reference 7 is not about differentiating young people practising various sports through quantitative indicators of body stability. The authors need to revise this.
Line 52 – also reference 8, is not about the impact of sports on balance. The authors need to revise this.
Line 133- how many trials were performed for each participant? Moreover, 20 seconds is quite a short period to evaluate balance, particularly with this population.
Lines 140 – 143 – Still not clear how the authors computed the SI. Moreover, reference 35 does not provide any information on this variable. The authors need to improve this part.
Lines 164 – Why did the authors introduce this type of analysis (Dunn’s multiple comparison test for pairwise comparison)? I don’t see the need to duplicate the statistical analysis (parametric and non-parametric). The authors should keep only the one that is the more appropriate for the type of data.
Conclusions” Research on the stabilometric platform among 352 young people can be used to select people with inborn better balance which can be pre- 353 dispositions to train archery”, has previously mentioned; this study does not allow the authors cannot assume this type of conclusion. The authors need to revise.
Author Response
Dear Reviewer,
Below I'm sending answers to your comments. Thank you for your valuable suggestions.
Abstract – As previously mentioned, the conclusions are speculative since the experimental design does not allow such conclusions. Therefore, the authors need to revise this.
We revised.
Line 49 – as far as I could understand, reference 7 is not about differentiating young people practising various sports through quantitative indicators of body stability. The authors need to revise this.
We revised.
Line 52 – also reference 8, is not about the impact of sports on balance. The authors need to revise this.
We revised.
Line 133- how many trials were performed for each participant? Moreover, 20 seconds is quite a short period to evaluate balance, particularly with this population.
Each participant were performed once. The research methodology and test time were adopted based on the literature and available recommendations regarding the duration of the balance test, ref. 42.
Lines 140 – 143 – Still not clear how the authors computed the SI. Moreover, reference 35 does not provide any information on this variable. The authors need to improve this part.
True. It should be ref. 43. We revised.
Lines 164 – Why did the authors introduce this type of analysis (Dunn’s multiple comparison test for pairwise comparison)? I don’t see the need to duplicate the statistical analysis (parametric and non-parametric). The authors should keep only the one that is the more appropriate for the type of data.
We used the Kruskal-Wallis Anova test in our analyzes. After the recommendation of one of the reviewers, we added Dunn's test, because it allowed for precise determination of the groups between which statistically significant differences were determined.
Conclusions” Research on the stabilometric platform among 352 young people can be used to select people with inborn better balance which can be pre- 353 dispositions to train archery”, has previously mentioned; this study does not allow the authors cannot assume this type of conclusion. The authors need to revise.
We revised it.
With Kind Regards
This manuscript is a resubmission of an earlier submission. The following is a list of the peer review reports and author responses from that submission.
Round 1
Reviewer 1 Report
The effect of various types of physical activity on balance and symmetry of lower limbs loading in young athletes
General Comments:
The writing in a lot of areas is anecdotal, and extremely vague. These do not strengthen the argument of the paper scientifically. A lot of these needs to be improved throughout the manuscript. I have outline some of these statements under specific comments. Some of the statements also make claims with no references cited.
There is absolutely no argument on why BMI, balance and symmetry were chosen to be assessed. Why are these variables important? What has been previously reported in literature in these areas?
I still have no clue how long the training was? Was it 3 years? Or 3 times a week and then “OR” is mentioned for 4.5-6 hours weekly. For how many weeks?
When the measurements actually recorded?
Very hard to generalize results to regular training, but results can be mentioned as BMI, balance and symmetry among soccer players, dancers, and swimmers.
I highly suggest the authors write this as an assessment paper of adolescents who played the three different sports and discuss the findings in that aspect. Not as a specific sport as an intervention, especially with not a clear idea when these measurements were actually made. Re-write the introduction specific to the variables assessed and how they are impacted by the specific sport played.
Specific Comments:
Abstract:
- BMI needs to be written full.
- Please mention participant demographics such as average age, height, weight, gender etc.
- Was 3 years the duration of the training?
- How long was the training specifically during the three years? How many times per week? All throughout the year for 3 years?
- Line 17 is not a reasonable claim given the methodology.
Introduction:
- Line 23 – I don’t think the word except is correct in this context, Sport, in addition to, is better
- Line 27 Typo
- Lines 32-34: This is purely anecdotal unless a reference is provided.
- Line 36: Highly suggest to avoid these types of statements. What transformation?
- Lines 43-45 another example of such writing.
- Line 50, 1 should be one. Anything below 9 write alphabetically.
- Lines 54-56: Reference?
- The purpose of the study is not justified.
Methods:
Please see my general comments. When were the measures taken?
Discussion:
The discussion actually talks about postural control and balance, which was a variable measured. There are statements that don’t entirely support the findings that need to be fixed.
Author Response
Dear Reviewer,
Thank you for your time and comments on the manuscript. We have made changes as suggested.
I believe that with these changes, the article will strengthen the argument of the paper scientifically and will be a source knowledge for other readers.
With Kind Regards,
Michalina Czarnota

Reviewer 2 Report
Dear Authors,
The work "The effect of various types of physical activity on balance and symmetry of lower limbs loading in young athletes” has potential to be developed and considered for publication. However, at the present moment, I feel it needs several revisions.
Please check the following comments:
The entire manuscripts need English grammar revision from a native speaker/writer. For instance, check the first line of the abstract. Also, L17 “Swimming training can complement training”. Also, L30 “many various”
Abstract: Please define all abbreviations the 1st time they are mentioned.
what was the sex of participants? Please provide more details about the characteristics of participants.
L11 – “They were aged 14 – 17” - This information was already mentioned. Please remove it.
Please provide more details about methodology applied.
Please add data from statistical analysis.
L19 - This is not a conclusion from you study. Please reformulate.
Keywords: Please remove the words that are already in the title.
Introduction:
L23-24 – Reference is needed
L31 – At least one more reference is required.
L32-35 - Is this your opinion? My suggestion is to add at least a reference
L36-42 and L44-49 - More paragraphs without references. It does not fit in a scientific manuscript. Please check
Line 51 also lacks a reference
L52 “Eric G. Post et al among 2011” - this reference is not in the text
L54-56 – Again, references are needed.
L55 - According to the following study recommendation, the terms of “load” should be replaced by "intensity" in all sections. Please check the entire manuscript.
https://www.jsams.org/article/S1440-2440(21)00212-7/fulltext
L71-73 – The aim of the study is not clear and hypothesis of the work is missing.
Finally, a general comment about introduction is that it does not properly support the title of the paper. The authors present several information about injury or trauma or progression of pubertal events, but without specific details. In the end the sport in analysis were not addressed.
Materials and Methods
This section also needs to be improved. It does not properly explain when data was collected and how did the sports in study influence the data collection. It was not clear what was the aim of the study and what knowledge did the authors want to show for scientific community.
Statistical analysis
Text is in Italic. Please check guidelines of the journal.
Based on my comment, I suggest major revisions for this work before consideration for further revision.
I hope you understand.
Thank you and best regards
Author Response

(The authors gave the same response as above.)

Round 2
Reviewer 1 Report
The authors have addressed most of the comments. However, I urge authors to take one more pass in reviewing the manuscript and making sure that this manuscript is demonstrating the differences in balance, BMI, etc. in a sample of adolescents with different types of training.
Author Response
Dear Reviewer,
We take one more pass in reviewing the manuscript and we make sure that this manuscript is demonstrating the differences in balance, BMI, and symmetry of lower extremities load in a sample of adolescents with different types of training.
Thank you for you revision.
Kind Regards,
Michalina Czarnota
Reviewer 2 Report
Dear Authors,
The manuscript was improved but after the first round of revision I was expecting more clarity in terms of scientific background to support the aims of the study.
Please check the following comments:
L30-31 – What are those side effect? Please add them.
L35-36 – references are needed.
L42-48 - references are needed.
L70 – “balance training of balance”. This sentence needs to be rewritten.
L74 - If it is rarely described, there are some references which describe it. So please add a reference.
L80-85 – Again, references are needed.
L98-99 - I understand that these sports were chosen based on a convenient sample, but some rationale in introduction should support the use of these 3 different sports.
L99 – parameter word is repeated.
General comment on introduction - After reading introduction, I can't see what is the rationale to support this sport over the others in terms of balance and symmetry of lower limbs? Please consider this comment to improve clarity of your introduction.
L121-122 - this sentence can be removed because you already explain this information.
L126 – “This method was considered”. Please add the word “was”.
L161 - check the space between words.
L170 - space between words is needed
L181 - space between words is needed
Discussion - This section should start by remembering the aims of this study and by showing the mains results. Then, the discussion should be reorganized to support (or not) your main findings. After reading it, it is still to confusing. There are no justifications for the non-significant results that you found on BMI, balance, center of gravity and center of the pressure.
A paragraph with limitations and practical application of this study is missed.
Best regards
Author Response
Dear Reviewer,
Thank you for your valuable comments. We have revised the article and hope that the changes have improved the quality of the manuscript.
Kind Regards,
Michalina Czarnota

Round 3
Reviewer 2 Report
Dear Authors,
The manuscript seemed to be improved but it still does not allow its replicability. Considering my previous comments, the authors failed to answer or follow my suggestions. I could accept that if the authors provide any justification, but this was not the case. The authors respond to reviewers by saying: “We improved throughout the manuscript, or We add references to statements making claims”. This is not the right way to review and answer the comments.
Moreover, there are several English errors. An English native or expert should review the manuscript.
Please check the following comments:
Abstract – The abstract lacks statistical results. Not only the p-value, but correlation values should be added. L20 - Check the English writing. Poor English. This should be reviewed.
Introduction – L40-41 - Check the English writing. Poor English.
L66-68 – The authors addressed an important topic which is the puberty, however this was not considered for analysis. So why did mention this topic, if it was not addressed (not even in limitations)?
L72 - You mentioned studies, but then only one was cited.
L73 – What studies?
L76 and L78 – BMI was already defined. Thus, the acronymous should be used. Please check the entire manuscript.
L89 - What do you mean? What character?
L91 – Check the writing.
Again, why did soccer, swimming and dancing were chosen? Once more, the authors failed in supporting their choices. The hypothesis seems to be supported based on your own results, but it should be supported with the literature. This was not properly addressed.
General comment on introduction - After reading introduction, I can't see what is the rationale to support this sport over the others in terms of balance and symmetry of lower limbs? Please consider this comment to improve clarity of your introduction.
Methods
L101-103 - You should mention the type of your study and not a speculation about studies being interesting or not.
L102 - What studies?
L104-109 - I can't understand why did the authors used this paragraph in the design?
L123 - What do you mean? Some participants were accessed in October, others, and December and other in April? How did authors control the training effect? Was the baseline condition similar for all participants? Much more details should be given. through the present information, it is not possible to replicate the study
Discussion - This section should start by remembering the aims of this study and by showing the mains results. Then, the discussion should be reorganized to support (or not) your main findings. After reading it, I can see that the same issue remains. In addition, there are no justifications for the non-significant results that you found on BMI, balance, center of gravity and center of the pressure.
“The results of the research confirm the assumed thesis that among 269 all subjects, the best results among the majority of parameters describing the balance were 270 obtained by swimmers” - How can the authors say that, if this was not addressed or supported in the introduction?
A paragraph with limitations and practical application of this study is missed.
Therefore, I must recommend rejection and an encouragement for future submission after major revisions. I hope the author can understand my comments and use them to improve your paper.